# Can we do better? A qualitative study in the East of England investigating patient experience and acceptability of using the faecal immunochemical test in primary care

Claudia M Snudden [ID],[1] Natalia Calanzani [ID],[1,2] Stephanie Archer [ID],[1] Stephanie Honey [ID],[3] Merel M Pannebakker [ID],[1] Anissa Faher,[4] Aina Chang,[4] Willie Hamilton [ID],[5] Fiona M Walter [ID] [1,6]

**Correspondence to**
Dr Claudia M Snudden;
cs2048@medschl.cam.ac.uk

## ABSTRACT

**Objectives** The faecal immunochemical test (FIT) is increasingly used in UK primary care to triage patients presenting with symptoms and at different levels of colorectal cancer risk. Evidence is scarce on patients' views of using FIT in this context. We aimed to explore patients' care experience and acceptability of using FIT in primary care.

**Design** A qualitative semi-structured interview study. Interviews were conducted via Zoom between April and October 2020. Transcribed recordings were analysed using framework analysis.

**Setting** East of England general practices.

**Participants** Consenting patients (aged ≥40 years) who presented in primary care with possible symptoms of colorectal cancer, and for whom a FIT was requested, were recruited to the FIT-East study. Participants were purposively sampled for this qualitative substudy based on age, gender and FIT result.

**Results** 44 participants were interviewed with a mean age 61 years, and 25 (57%) being men: 8 (18%) received a positive FIT result. Three themes and seven subthemes were identified. Participants' familiarity with similar tests and perceived risk of cancer influenced test experience and acceptability. All participants were happy to do the FIT themselves and to recommend it to others. Most participants reported that the test was straightforward, although some considered it may be a challenge to others. However, test explanation by healthcare professionals was often limited. Furthermore, while some participants received their results quickly, many did not receive them at all with the common assumption that 'no news is good news'. For those with a negative result and persisting symptoms, there was uncertainty about any next steps.

**Conclusions** While FIT is acceptable to patients, elements of communication with patients by the healthcare system show potential for improvement. We suggest possible ways to improve the FIT experience, particularly regarding communication about the test and its results.

## STRENGTHS AND LIMITATIONS OF THIS STUDY

⇒ Interested participants were contacted within 4 weeks of performing a faecal immunochemical test, helping to reduce recall bias.

⇒ All participants were white (British or other) and English speakers—views may differ for those identifying with other ethnicities or who do not understand English well.

⇒ The use of Zoom for participant interviews may have hindered those unable to navigate the required technology from taking part.

## INTRODUCTION

Colorectal cancer (CRC) is the third most common cancer worldwide and incidence is increasing.[1] Globally, mortality rates are decreasing, in part due to increased screening, improved early diagnosis and advances in treatment.[2 3] Within the UK, the majority of patients diagnosed with CRC will first present to their general practitioner (GP) with abdominal symptoms.[4 5] GPs act as gatekeepers within the UK's National Health Service, deciding which patients require onwards referrals to secondary care, and how urgently.

Diagnosing CRC, particularly in its early stages, can be difficult as symptoms of CRC are broad and non-specific.[6] GPs in England currently follow National Institute for Health and Care Excellence (NICE) 2015 guidelines for patients with suspected cancer to identify higher risk patients who should be urgently referred for further investigation.[7] In 2017, NICE recommended the use of the faecal immunochemical test (FIT) within primary care to help triage low risk patients with possible CRC symptoms that do not meet

urgent referral criteria.[2 8] FIT can be used for both ruling in and ruling out CRC.[9 10] Alongside its use in symptomatic patients, FIT is also used in the UK's bowel cancer screening programme, having replaced the previously used guaiac faecal occult blood test.[11]

The threshold set to define a positive FIT result (ie, the haemoglobin (Hb) cut-off concentration) differs between populations. For symptomatic patients, NICE advises that those with a FIT result ≥10 μg Hb/g faeces are referred on for further diagnostic testing; of these about 7% will be diagnosed with CRC.[2 9] In contrast, England uses a cut-off of 120 μg Hb/g in asymptomatic screening to select who requires further investigation.[11] Variations in thresholds are due to the differences in risk between asymptomatic and symptomatic populations; the risk of cancer is higher for those with symptoms therefore the threshold for further investigations is lower.

As a result of the COVID-19 pandemic, FIT use in primary care was intensified to help prioritise urgent referrals.[12–15] In June 2022, The British Society of Gastroenterology released additional guidelines recommending the continued use of FIT as a diagnostic triage tool for all urgent referrals.[16 17] NICE also plans to publish additional guidance on FIT by November 2023.[18] Given the increasing reliance of FIT use in primary care, it is critical to understand what patients think of the test and whether they find it acceptable. Acceptability is defined as a multifaceted construct comprising affective attitude, burden, perceived effectiveness, ethicality, intervention coherence, opportunity costs and self-efficacy.[19] It is a necessary condition for the effectiveness of an intervention or test; from the patient perspective, if a test is considered acceptable, patients are more likely to adhere to the proposed investigation which in turn results in improved clinical outcomes.[19 20]

Studies conducted in screening contexts have shown that disgust associated with performing stool tests and certain procedural aspects of FIT can present barriers to uptake, however, the test is often found acceptable.[21–26] Nonetheless, there is still limited research into FIT experience and acceptability, particularly within the context of symptomatic patients.[27] Understanding both care experience and acceptability is vital to ensure good quality of care for patients.[28]

Since symptomatic populations have varied (and higher) levels of risk compared with an asymptomatic screening population, evidence on the patients' experience of care when and after doing FIT is crucial. To our knowledge, only two other studies have reported on symptomatic patients' views of FIT; one quantitative study focusing on usability and acceptability,[27] and one mixed-methods study evaluating patient experience and satisfaction.[29]

Therefore, this study aims to build on existing literature, providing an in-depth qualitative exploration of both FIT experience and acceptability for patients presenting to primary care with possible symptoms of CRC.

## METHODS

### Design, setting and population

This qualitative substudy was undertaken as part of the FIT-East study, which was set across general practices in the East of England Cancer Alliance (Suffolk & North East Essex and Norfolk & Waveney).[30] FIT-East included 507 patients aged 40 years and older with possible symptoms of CRC, but who did not immediately meet urgent referral criteria, for whom a FIT was requested by the GP.[30] FIT kits contained a study recruitment letter and consent form; participants could tick a box if interested in taking part in interviews.

We undertook semi-structured interviews with patients to investigate experience and acceptability of using FIT in primary care. Interviews were also carried out with GPs (reported elsewhere[31]). We followed the Consolidated Criteria for Reporting Qualitative studies guidelines.[32]

### Sampling

We undertook purposive sampling based on age, gender and FIT result to obtain a range of experiences among patients who consented to take part in the FIT-East study. Potential participants were contacted by email (provided by patients in the consent forms). To reduce recall bias, we contacted patients within 4 weeks of FIT being sent to the laboratory.

Ninety-nine participants responded positively to the initial email and were invited for an interview. Additional informed consent was required and obtained for all patients. Interviews were carried out until no new topics were discussed: 45 participants were interviewed, mean duration 39.5 min. Due to the COVID-19 pandemic, interviews were carried out remotely using the Zoom platform and audio recorded. One interview could not be analysed due to a technical fault during recording; therefore 44 participants were included. Each participant was individually interviewed once by one or two experienced female health services researchers with a background in psychology (SA and MMP) or nursing (SH).

### Interview topic guide

The guide (online supplemental file 1) explored patient experiences of being asked to undertake a FIT in primary care, covering areas including: obtaining the test, using it, returning it and receiving the results. The topic guide was piloted by the research team before use. In order to better understand patients' experiences during the COVID-19 pandemic, additional questions were added in June 2020. Interviews were carried out between April and October 2020.

### Data analysis

Verbatim transcripts were checked, anonymised and analysed using framework analysis.[33] This allows for both inductive and deductive approaches. An inductive approach was adopted for data analysis, influenced by our research question and specific definitions of acceptability.[33]

## Box 1 Theoretical models

Three theoretical models, listed below, were used to help inform and conceptualise the themes during data analysis. These models were chosen to underpin important complementary but distinctive aspects of the cancer diagnostic pathway[36 37] including the role of patient factors/characteristics,[36 37] the construct of acceptability[19] and patient's experience of care.[28]

1. The Model of Pathways to Treatment[36 37] describes events, processes, intervals and contributing factors (such as patient experience) underpinning the pathway to diagnosing cancer for patients presenting with symptoms.
2. The Theoretical Framework of Acceptability[19] describes seven constructs of acceptability: affective attitude, burden, perceived effectiveness, ethicality, intervention coherence, opportunity costs and self-efficacy.
3. Forster's *et al* considerations when assessing acceptability of diagnostic tests[28] describe dimensions such as patient-centred care, continuity and coordination of care and waiting times.

Four researchers (one academic GP (CS), one health services researcher (NC) and two medical students (AF and AC)) repeatedly read the transcripts and coded the interviews. Eight interviews were initially coded inductively by CS to discover potentially unexpected aspects of the participants' experiences. An analytical framework was then developed using codes from the eight initial transcripts; codes were grouped into categories, influenced by the relevant aspects of three theoretical models (box 1). The framework was then used to systematically index the remaining transcripts, including double coding of a quarter of the data set. Codes and categories were constantly refined during analysis. Data were charted into a series of matrices from which themes were developed. Consistency of coding was discussed regularly during team meetings; themes were also amended with guidance from senior team members (SA and FMW). NVivo V.12 was used to facilitate analysis.[34]

Participant age (range), gender (M for male and F for female) and FIT results (positive, negative or unclear) are reported alongside quotes to aid interpretation of findings. Quotes were chosen to illustrate findings while also ensuring a good spread of participants and their characteristics.

### Patient and public involvement

Patient-public partners from the CanTest Collaborative[35] (responsible for FIT-East) provided feedback on the original and amended interview topic guides. Representatives also read and commented on interview transcripts; these comments were considered when iteratively revising our analysis.

### RESULTS

Table 1 describes characteristics of the included participants and describes how FIT results were received. About one-fifth (18%) had a positive FIT result. Outcome data

**Table 1** Demographic characteristics and FIT results (n=44)

| Characteristic | Variable | N | % |
|---|---|---|---|
| Sex | Female | 19 | 43 |
| | Male | 25 | 57 |
| Age (years) | 40–49 | 3 | 7 |
| | 50–59 | 17 | 39 |
| | 60–69 | 10 | 23 |
| | 70–79 | 11 | 25 |
| | 80–89 | 3 | 7 |
| FIT result | Negative | 35 | 80 |
| | Positive | 8 | 18 |
| | Unclear* | 1 | 2 |
| How FIT result was received | Patient called practice† | 12 | 27 |
| | Patient did not receive results/is unsure‡ | 11 | 25 |
| | Practice called patient§ | 9 | 20 |
| | Patient told face-to-face¶ | 4 | 9 |
| | Patient received letter** | 3 | 7 |
| | Patient checked results online** | 2 | 5 |
| | Other†† | 3 | 7 |
| Education level | None | 3 | 7 |
| | GCSE (or equivalent) | 4 | 9 |
| | A level (or equivalent) | 5 | 11 |
| | Degree (or equivalent) | 18 | 41 |
| | Other higher education | 10 | 23 |
| | Missing | 4 | 9 |
| Ethnicity | White British | 35 | 80 |
| | White Irish | 1 | 2 |
| | Any other white background | 2 | 4 |
| | Missing | 6 | 14 |
| Indices of Multiple Deprivation quintiles | 1 (most deprived) | 0 | 0 |
| | 2 | 4 | 9 |
| | 3 | 7 | 16 |
| | 4 | 4 | 9 |
| | 5 (least deprived) | 6 | 14 |
| | Missing | 23 | 52 |

*One patient described doing the FIT in detail, but on the patient database FIT was recorded as 'not done'. As the patient clearly had done the test, this was recorded as an unclear result.
†One positive result (patient reported this as inconclusive during the interview).
‡Did not receive (n=7, all negative), unsure about it (n=4, two negative, one positive and one missing).
§GP (n=6, two positive results), nurse practitioner (n=1, negative—nurse called about something else and the patient asked for results); receptionist (n=2, one positive result).
¶GP (n=2, all negative), nurse (n=1, positive), receptionist (n=1, negative).
** All negative results.
††Cannot remember how results were received (n=1, positive), GP contacted but unclear if by phone or face-to-face (n=1, negative), over the phone but unclear who called whom (n=1, positive).
FIT, faecal immunochemical test; GCSE, General Certificate of Secondary Education; GP, general practitioner.

for whether participants received a cancer diagnosis was not available.

### Overview and context

We identified three themes and seven subthemes describing patient experience and acceptability of using

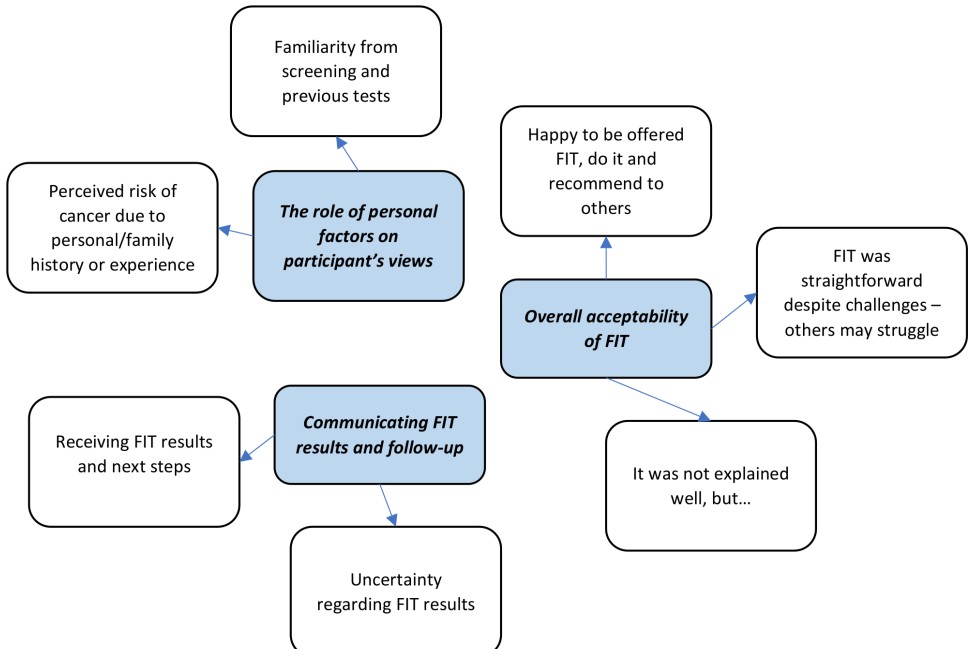

**Figure 1** Themes and subthemes: patient experience and acceptability of using FIT. FIT, faecal immunochemical test.

FIT after presenting in primary care with possible symptoms of CRC (but not meeting immediate criteria for urgent referral) (figure 1).

As the study was performed during the COVID-19 pandemic a brief description of COVID-19-related themes is available for context (online supplemental file 2). The pandemic did not appear to influence participants' experiences of using FIT, but it did affect how participants accessed their GP prior to receiving the test.

### The role of personal factors on participant's views
#### Perceived risk of cancer due to personal/family history or experience

A personal history of having had cancer in the past encouraged participants to seek help quickly for their symptoms. Similarly, cancer experiences from family and friends influenced their decision to seek help after noticing bodily changes, and to do the test when it was offered:

> My father has had bowel cancer. He basically had it once and they took a section of the colon out. After a few years it returned so they removed the whole of his colon and he's now been…you know, God bless he's still alive and it's been sort of over 15 years he's been clear but there is that sort of history that I was aware of. So I thought, alright, better…you know, because it had changed slightly, yes, and it was bleeding more often, I thought I'd better go and get it looked at. (PT18, M, 50–59 years, negative)

Participants became concerned that their symptoms could be a sign of bowel cancer—either from their own knowledge of 'red flag' symptoms or from reading about symptoms online, leading them to actively seek advice. Others became increasingly anxious or concerned about the severity of their symptoms, particularly if things had worsened over time or were causing significant discomfort.

On the other hand, having had a relatively recent examination or negative investigation (such as colonoscopy) often reassured them there was nothing serious and being offered a FIT was not considered to be worrying or concerning.

#### Familiarity from screening and previous tests
Participants were familiar with doing stool tests, usually as part of bowel screening or previous diagnostic processes. As a result, they often displayed no concerns with the idea of doing FIT:

> Well, I've been doing them for years. It didn't upset me, if that's what you mean. It's just one more thing, going to the bog isn't it? Going to the loo? Except this time you put it on a piece of […] paper, and get a smear and stick it in the packet, and then seal it up. (PT1, M, 70–79 years, negative)

Participants who had had previous experiences within secondary care (such as colonoscopies due to diverticular disease) described the FIT experience as less daunting:

> I was quite relaxed about [being asked to do FIT] because I'd had a colonoscopy before, that was 5 years ago, so that was more nerve-racking. (PT25, F, 60–69 years, negative)

### Overall acceptability of FIT
#### Happy to be offered FIT, do it and recommend it to others
Overall, participants reported they were glad or relieved to be offered FIT and accepted the need to do it. This was true even when the potential unpleasantness or difficulties of doing a FIT were recognised. For some, doing a

test could lead to an 'earlier diagnosis' and the potential to start treatment sooner if there was something serious going on. For others, acceptance was associated with wanting to find an answer to their symptoms, especially if these had been particularly distressing:

> I guess I felt a little bit strange, kind of, like here I am, you know, lining the toilet to catch my stools. But when you know that it's something that you want to get done and want to get answers for you just get on and do it really. (PT12, F, 40–49 years, negative)

When reflecting on why *others* might not want to do FIT, common reasons included the fear of a potential cancer diagnosis and finding the process disgusting or embarrassing:

> Probably because it's not a very nice thing to have to do, you know. Playing around with poo is not something that everybody thinks, 'oh that sounds like fun'. (PT3, F, 60–69 years, negative)

Every participant reported being happy to repeat FIT if necessary, and all would willingly recommend FIT to others if they needed it.

### FIT was straightforward despite challenges—others may struggle

Participants commented on challenges they faced when physically collecting the FIT sample: often these were difficulties writing the label, 'catching' the stool sample without it falling into the toilet water or when suffering with diarrhoea, and finding the FIT stick small or 'very fiddly' to handle.

Anxiety was also expressed regarding taking the FIT sample and 'wanting to do it right'. Furthermore, participants reported wanting to get the sample done and feeling relieved after completing the test:

> I wanted to get it out of the way, it's not exactly difficult, but it was just the having to, you know, deal with it. So I woke up thinking, 'oh, here we go, you know, I need to do this' and when I'd got things in the specimen jars and I sorted myself out, I was glad it was over. (PT23, F, 50–59 years, negative)

Methods to overcome difficulties collecting the sample were also described, drawing on prior experiences when completing bowel cancer screening tests. Nonetheless, participants (even those reporting these challenges) were confident about their ability to do the test, often remarking how simple and 'straightforward' FIT was.

In particular, participants found the instructions generally easy to follow. Nonetheless, they discussed why *others* might struggle with FIT—from potential difficulties faced by having arthritic or shaky hands, poor eyesight or an upset stomach, to issues following instructions.

### It was not explained well, but…

Participants agreed that the explanations given by their healthcare professional (usually a GP or practice nurse) regarding what FIT was and why it was indicated, tended to be poor:

> My GP didn't, actually, tell me what it was. I had to Google it to even find out what it was. He said 'we'll leave you out a FIT test', and I had no idea what it was. (PT14, M, 60–69 years, positive)

Nonetheless, some defended the professionals in such situations, attributing the limited information to common constraints in a consultation such as limited time to go through different medical problems.

Often participants were asked to do multiple stool tests as part of their symptom work-up, leading to confusion regarding the differences between tests. Frequent references were made to being handed a bag containing a FIT and being told to just follow the instructions. For two participants, the poor communication even resulted in confusion of the test with the similarly named Fitbit watch.

While the text explanation was often inadequate, participants had very good intuition about linking FIT to CRC. Participants reported that GPs alluded to cancer but did not use the word 'cancer' specifically, choosing instead to describe it as 'something sinister' or 'something nasty lurking'. Despite this, many appeared to interpret the GP's language appropriately:

> When he suggested doing a FIT test, which he explained to me what it was, I thought 'yeah, it makes sense really because I know what you're after'. He didn't tell me what he was after, but I thought 'I know what you're looking for'. (PT24, M, 50–59 years, negative)

Participants' understanding of how FIT worked varied considerably. The most common explanations focused around looking for traces of (often non-visible) blood in the stool, but others described FIT as 'a bit of an MOT check', being used to 'determine the health of the gut', 'check the digestive system is working properly' or look for 'abnormalities within the poo'. The internet was sometimes used to further research FIT and causes of blood in the stool.

While most participants demonstrated good intuition or a basic understanding of FIT, some had no awareness as to why they were asked to do FIT, yet did it anyway:

> I just got given the test to do and, I suppose, in a roundabout, sort of, way, she was saying, you know, 'you need to have the test done', but I didn't understand it and I was puzzled about it […] I think that's probably why I kept reading the notes about it, because I kept thinking, 'well, I'm not sure why I'm doing this'. (PT33, F, 50–59 years, negative)

Participants also talked about trusting the GP and their advice, and how this influenced their decision to do the FIT, how it would affect their decision to do it again or recommend it to others:

I like her, I think she's an excellent GP. If she asks me to do something, to participate or provide a test for example, I trust her to be asking that for a good reason and I will instinctively comply with any requests for tests or anything else as I'm assuming that it's going to provide information, test results or whatever which will be for my direct benefit. (PT19, M, 60–69 years, negative)

### Communicating FIT results and follow-up
#### Uncertainty regarding FIT results

Participants reported contrasting attitudes while waiting for test results. Some discussed the anxiety surrounding the waiting period, at times hypothesising what would happen next in the worst-case scenario.

In contrast, others were unaffected by waiting, seemingly managing to 'forget about it' and carry on with life as usual. This was often linked to either normalising FIT to a more 'standard' test 'just like having a blood test', or expecting to receive a negative result and therefore not seeing reason to worry.

Participants felt they were not given clear information by the GP on when and how to access their results. They suggested that chasing results should be practice-led rather than patient-led, regardless of the result. Some participants did not receive their results at all, others contacted the practice for these, and others assumed that the practice would only contact them if there was a positive result and reported not being worried as 'no news is good news':

I haven't actually been told anything so I presume they were negative or there were no concerns because no GP has contacted me. (PT20, F, 50–59 years, negative)

#### Receiving FIT results and next steps

For those who received their test results, there were often comments on receiving these quickly. Nonetheless, the way results were communicated was not always well received, both for positive and negative results:

The receptionist at the GP surgery telephoned me and said the doctor had referred me[…] and I said, 'oh, why is that then', and she…she didn't seem to know and I said 'I just had a FIT test, what was the result of that'? Pause, and she said, 'oh, yes, it was positive. I expect that's why it is'. And I'm not joking, that's exactly how the conversation went. (PT14, M, 60–69 years, positive)

I think one slightly disappointing aspect I suppose is that when I phoned for results, I only spoke to the receptionist, and all they were able to tell me was no further action necessary. And I kind of thought, 'oh well, it's great in one way and I was pleased with that, but it kind of didn't explain why I'd had problems earlier' and I thought, 'oh, do I just wait to see if I

ever have this problem again'. (PT41, M, 50–59 years, negative)

For participants with a positive result, there were reported delays in having a colonoscopy, particularly during the COVID-19 pandemic. Participants were often stoical about this situation. For those with a negative result, a sense of relief and reassurance was frequently reported. However, when symptoms persisted, there was uncertainty regarding what to do next:

No, I don't know if there's a follow-up or anything, if you detect nothing does it just get swept under the carpet or do I stay on your database and do you check up again or…I don't know? (PT6, M, 60–69 years, negative)

Some said they would have valued a follow-up consultation with their GP, but in practice this rarely happened, leaving participants feeling their problems had not been fully resolved. Others believed that the GP would not be able to offer them anything further even if they did arrange a further appointment.

### DISCUSSION
#### Summary

This study of patients presenting in primary care with symptoms of possible CRC but not immediately meeting criteria for urgent referral found FIT an acceptable and straightforward test, despite some challenges manipulating the test kit. Participant characteristics such as past test experiences and perceived personal risk of cancer influenced attitudes towards FIT. Participants were also happy to do FIT again and recommend it to others, and trusted that their GP knew what was best for them. On the other hand, test explanations from health professionals were generally poor. This could be due, in part, to the lack of clarity surrounding FIT implementation and diagnostic pathways within primary care during the COVID-19 pandemic. There was also reported dissatisfaction with the way results were received, for both positive and negative results. Many participants did not receive results at all, and several assumed that 'no news is good news'. For some, there was no follow-up after a negative result, even when symptoms persisted.

#### Strengths and limitations

To our knowledge, this is the first qualitative study to explore experience and acceptability of FIT for symptomatic patients. We provide rich data on patients' care experiences and acceptability of FIT use in primary care. With the increased use of FIT in this setting, it is essential to understand patient views on the test and identify areas where care needs to be improved. Three well-established theoretical models were used to guide conceptualisation of the themes during analysis (box 1).[19 28 36 37] We found that the models complemented each other, with overlap between different elements. Patient factors[36] such as

perception of risk, previous knowledge and experience influenced test acceptability,[19] including the perceived burden, understanding why the test was being done (intervention coherence) and perceived ability to do the test (self-efficacy). Conversely, the overall test acceptability contrasted with the often reported less than ideal experience of care[28] before, during and after doing the test. Using a single model alone would have resulted in missing several nuances regarding acceptability, the role of patient factors and the quality of the experienced care.

Interested participants were contacted within 4 weeks of performing FIT, helping to reduce recall bias. There was good variation among participants in terms of age, gender and FIT result. However, only a sample of patients who took part in the FIT-East study were interviewed; views of those who did not participate in the interviews may differ. The sample was also limited to the East of England. All participants identified themselves as white, slightly higher than the proportion reported by the UK census for this region,[38] and understood English well. Experiences of patients identifying with other ethnicities, and of those who do not speak English well may differ and need to be further investigated. Limited evidence on FIT use in symptomatic patients indicates that ethnicity may influence patient willingness to do the test.[27] Finally, we were not able to explore variations in views based on different levels of socioeconomic deprivation due to large amounts of missing data (52%). The relationship between poor care experience and test acceptability may differ for these groups; different levels of knowledge and experience may also play a role on FIT acceptability.

Due to resulting restrictions from the pandemic, interviews were conducted over Zoom, hindering those unable to navigate the required technology from taking part. No patients aged under 40 were recruited (as defined by our inclusion criteria). As FIT becomes more widely used in primary care, it will be key to explore the attitudes of this group, less familiar with stool testing and without involvement in prior screening programmes—particularly as the incidence of CRC is increasing in the under 50s.[39–41]

### Comparison with existing literature
Consistent with existing literature (although mostly investigating asymptomatic populations), FIT is seen as relatively straightforward with clear instructions.[23–25 27] Nonetheless, similar challenges to collecting the sample were reported including difficulties writing the label and handling the FIT stick.[24 26] Familiarity with stool tests is associated with higher intentions to use related tests in the future, aligning with the confidence displayed by participants with experience of screening.[27] Social influences have also been associated with encouragement of FIT uptake, impacting an individual's perceived risk of, or susceptibility to, cancer.[23 42]

Participants displayed different levels of proactivity in seeking FIT results, reflective of behaviours seen in the literature.[43] Patients reported wanting further clarification on how results are communicated; existing uncertainties

---

**Box 2 Recommendations for research, practice and policy**

**Research**
⇒ Investigate (1) challenges in collecting the stool sample, particularly for older people and those with mobility issues; and (2) improvements to sample collection (such as providing a collection container) and instructions (such as clarification on what to do in the case of diarrhoea).
⇒ Investigate whether patients' presenting symptoms (high vs low risk) influence their perception of the faecal immunochemical test (FIT), as we found that perceived risk and susceptibility influenced acceptability.
⇒ Seek to understand the views of patients who choose not to complete FIT after being offered it, to determine (1) why FIT may not be acceptable to some and (2) whether reasons align with issues identified in this study (eg, test burden and limited information).
⇒ Investigate the views of those living in areas of higher deprivation, and those identifying with different ethnic groups other than white.

**Practice**
⇒ Work to improve general practitioner (GP) communication with patients regarding why FIT is being done, what it looks for, and what a positive result means. Clear communication can also concurrently reduce patient confusion when multiple tests are requested simultaneously.
⇒ Consider implementing a specific time slot for calling for results and promote these changes to patients through information posters around the surgery and batch SMS message updates.
⇒ Specify requirements for improved training of reception staff in handling results. This would include guidance on communicating potentially sensitive information.
⇒ Capitalise on the growth in use of patient messaging platforms such as Accurx,[47] resulting from the COVID-19 pandemic, to develop formal pathways within general practices for proactively communicating negative results and safety-netting advice:
  ⇒ Encourage direct SMS messaging of negative results to patients using pre-built templates which also contain information advising patients to contact their GP again if symptoms persist.
  ⇒ For those without a recorded mobile phone number, adopt a receptionist-led system of contacting the patient's landline, sending a letter informing them of their negative result if unable to get through.

Policy
⇒ Instigate policy nationally to support standardisation of practices, such as those suggested above, within primary care.
⇒ Ensure patients are included within the design and evaluation of future services and care models.

---

were anxiety-inducing at times.[24 43 44] To our knowledge, there is no other available evidence on follow-up (or lack thereof) of patients with persisting symptoms and a negative FIT result.

### Implications for research, practice and policy
Building on our findings and published literature, alongside considerations on what is feasible within the UK context, we have developed recommendations for research, practice and policy to improve both the FIT experience and patient safety (box 2). Rationale for these recommendations is described below.

While FIT acceptance was high, there are still opportunities for improvement, particularly regarding processes to collect the stool sample, and better understanding FIT acceptability across different population groups.

Reports of questionable care about explaining the test, providing results and following-up are also concerning.[45] While participants still found FIT acceptable and trusted that the GP knew what was good for them, care experiences caused anxiety to patients and indicate risks regarding continuity of care. Time waiting for results did not seem to be a problem for those who got them; but many reported that they never received the results. While no news was indeed often good news, some patients who chased their results found out they were positive. It is vital that GPs are clearer about when results are expected and how to access them, and safety-netting measures are in place to avoid missing cancers and provide better support to patients.

Finally, our results can support policymakers regarding FIT use and implementation within primary care. While flexibility and adaptability are required in the provision of services so different needs can be met,[46] it is vital that national policies define standards on acceptable levels of service, particularly regarding how and when information on test results are provided.

## Conclusion

Overall, patients presenting in primary care with symptoms of possible CRC find FIT an acceptable test, even when facing challenges to collect their samples. Acceptability was influenced by personal knowledge and previous experience. FIT acceptability was high despite reports of poor experience of care, particularly regarding communication about the test and its results. There is scope for improving the test itself and its communication to enhance patient experience. Future studies should investigate patient acceptability and experience of care among those who choose not to do FIT, across different ethnic groups and different levels of social deprivation.

**Author affiliations**
[1]Department of Public Health and Primary Care, University of Cambridge, Cambridge, UK
[2]Academic Primary Care, University of Aberdeen Institute of Applied Health Sciences, Aberdeen, UK
[3]Leeds Institute of Health Sciences, University of Leeds, Leeds, UK
[4]School of Clinical Medicine, University of Cambridge, Cambridge, UK
[5]University of Exeter Medical School, University of Exeter, Exeter, UK
[6]Wolfson Institute of Population Health, Queen Mary University of London, London, UK

**Acknowledgements** We thank all the patients who agreed to be interviewed. We thank the North East Essex and Suffolk Pathology Services at the East Suffolk and North Essex NHS Foundation Trust, and the Clinical Biochemistry section of Laboratory Medicine at Norfolk and Norwich University Hospitals NHS Foundation Trust. They have added a study recruitment letter, consent form and prepaid envelope to each FIT kit, distributed these to GP practices and provided data on FIT results. We thank Professor Niek de Wit and Dr Peter Holloway for their support in the earlier stages of the study. We are grateful to Marije van Melle for her work in the early stages of study design. We thank James Brimicombe for data management advice, support and expertise, and Andy Cowan for his help collecting and managing FIT-East study data. We are grateful to the CanTest Patient & Public Involvement (PPI) Panel (special thanks to Margaret Johnson) for reading and commenting on interview transcripts.

**Contributors** MMP and FMW planned the original study; WH and FMW secured funding. SH, SA and MMP carried out the interviews. Analysis was planned and carried out by CMS, NC, SH, SA, AF, AC and FMW. CMS and NC drafted the manuscript with senior input from SA and FMW and revised it after critical reviews from all authors. All authors read and approved the final manuscript. CMS is acting as the guarantor.

**Funding** This work was supported by the CanTest Collaborative, which is funded by Cancer Research UK grant number C8640/A23385, of which FMW and WH are co-Directors, and NC, MMP and SH are postdoctoral researchers. This work was co-funded by the National Institute for Health Research (NIHR) Policy Research Programme, conducted through the Policy Research Unit in Cancer Awareness, Screening and Early Diagnosis, PR-PRU-1217-21601. CMS is an academic clinical fellow funded by the NIHR. The views expressed in this publication are those of the authors and not necessarily those of the National Health Service, the NIHR or the Department of Health. The funding sources had no role in the study design, data collection, data analysis, data interpretation, writing of the report or in the decision to submit for publication.

**Competing interests** None declared.

**Patient and public involvement** Patients and/or the public were involved in the design, or conduct, or reporting, or dissemination plans of this research. Refer to the Methods section for further details.

**Patient consent for publication** Not applicable.

**Ethics approval** Ethical approval was granted by the East of England Cambridge Central Research Ethics Committee – reference 19/EE/0036.

**Provenance and peer review** Not commissioned; externally peer reviewed.

**Data availability statement** Data are available upon reasonable request.

**ORCID iDs**
Claudia M Snudden http://orcid.org/0000-0002-7355-4317
Natalia Calanzani http://orcid.org/0000-0002-5068-2543
Stephanie Archer http://orcid.org/0000-0003-1349-7178
Stephanie Honey http://orcid.org/0000-0002-2675-7824
Merel M Pannebakker http://orcid.org/0000-0003-2918-0570
Willie Hamilton http://orcid.org/0000-0003-1611-1373
Fiona M Walter http://orcid.org/0000-0002-7191-6476

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
