## [Reviewer comments · BMJ Open]

ARTICLE DETAILS

TITLE (PROVISIONAL)	Can we do better? A qualitative study in the East of England investigating patient experience and acceptability of using the faecal immunochemical test in primary care
AUTHORS	Snudden, Claudia; Calanzani, Natalia; Archer, Stephanie; Honey, Stephanie; Pannebakker, Merel; Faher, Anissa; Chang, Aina; Hamilton, Willie; Walter, Fiona M

VERSION 1 – REVIEW

REVIEWER	Jervaeus, Anna Karolinska Inst
REVIEW RETURNED	07-Mar-2023

GENERAL COMMENTS	Thank you for letting me review this manuscript titled: It is an interesting paper and I find similarities to the research we have conducted here in Sweden. Can we do better? A qualitative study investigating patient experience and acceptability of using the faecal immunochemical test in primary care Background Relevant to reflect upon: Are the relevant concepts and aspects of the study defined here? For whom are you writing this? Do you need to include any more information, for instance regarding colorectal cancer? I would suggest to expand the background based on these questions. What does NICE stand for? Please explain abbreviations throughout the manuscript. It would be good to put describe the health care system in the UK. Help the reader to grasp the context both regarding health care sector and this part of England. Who do you want to address this paper to? I would suggest formulating a clearer rationale. Do we need more of this type of research and why? What is new? Aim The aim is to explore FIT experience and acceptability. How is acceptability defined? How is acceptability addressed in the interview guide? Please see further under Methods. Methods It's good that COREQ is acknowledged. My major concern here is that a deductive analysis seems to have been conducted. That is not, however, visible in the design description. It is not clear how the analysis process has been conducted so I would like to see a more detailed description. It could be in the format of a visual presentation about the process, adding on to the text. Results
--

	“Many”, “several”, “most” participants are frequently used. Is this relevant for a qualitative paper? Isn't it the variation that should be in focus? I would suggest decreasing number of citations in the result section. My impression now is that the section is very much a description of the different citations and that is not my view on how a qualitative result should be structured. How is the deductive perspective visible here? Discussion It is stated here that three well-established theoretical models were used but I don't see how. The authors state that the models complemented each other but how is that shown to the reader? Please see under method. What about the section on COVID? How is it going to be addressed? Do you have such solid evidence as to be able to come up with all those recommendations or are they based on other research as well? I miss a more detailed discussion regarding strengths and limitations. I miss a Conclusion.
--	---

REVIEWER	Nepogodiev, Dmitri University of Birmingham, Academic Department of Surgery
REVIEW RETURNED	17-Mar-2023

GENERAL COMMENTS	This is a well executed study that has been transparently reported with clear recommendations for clinical practice. Unfortunately, it is not possible to address the study's main limitation, which is that the patients interviewed were all white and disproportionately had degree-level qualifications and higher socioeconomic status. It is well documented that bowel screening uptake is significantly lower in populations with lower socioeconomic status and in ethnic minorities, so it is a missed opportunity to not have included a more diverse patient sample to explore potential inequalities within symptomatic patient groups. Could the authors comment on what they did find in terms of differences between socioeconomic groups? The need for further research in to socioeconomic and ethnic group inequalities using both qualitative and quantitative methods should be highlighted as a future research priority (box 1). Please add to main manuscript that you have used COREQ.
--

VERSION 1 – AUTHOR RESPONSE

Reviewer comments	Author response
Reviewer 1	
Aim The aim is to explore FIT experience and acceptability. This is still not addressed in connection to the aim: How is acceptability defined? How is acceptability addressed in the interview guide? Please see further under Methods	Following our previous revisions, we have now moved our definition of acceptability into the Introduction (paragraph 4). “Acceptability is defined as a multi-faceted construct comprising affective attitude, burden, perceived effectiveness, ethicality, intervention coherence, opportunity costs, and self-efficacy¹⁹”

	The questions in the interview topic guide were broad to encompass constructs within Sekhon et al's theoretical framework of acceptability (TFA). We have added our response here again for information: Within the interview topic guide, questions were broad to both encompass constructs of Sekhon's TFA, whilst also leaving scope for participants to talk about other issues regarding FIT that they felt were important. For example, affective attitude was investigated by asking questions such as "How did you feel when the GP suggested having a FIT-poo test?" and "How did you feel following the test?". Burden was explored through questions such as "How did you feel about doing the test – physically, emotionally?". Intervention coherence was investigated by asking "How would you explain the test to someone else?" The question "How was the FIT-poo test introduced and explained" was designed to shed light on both the care experience and the construct of perceived effectiveness. We did not have questions focusing specifically on ethicality and opportunity costs – while answers to open-ended questions could have still approached issues relevant to these, we found that these constructs were less relevant to the participants' experiences.
Methods My major concern here is that a deductive analysis seems to have been conducted. That is not, however, visible in the design description. It is not clear how the analysis process has been conducted so I would like to see a more detailed description. It could be in the format of a visual presentation about the process, adding on to the text. New comment: This is still a concern to me, what is the design of this study? To me, a qualitative study is not enough as a description of the design. I would also like to see a rationale for the use of Acceptability much earlier in the manuscript, preferably in the background. Why is this concept relevant and what is included in these conceptual models.	Regarding addressing the rationale for the use of acceptability earlier on in the manuscript, we have now added a definition to the Introduction, see above. Furthermore, we have briefly explained why the concept is relevant (Introduction, paragraph 4): "It is a necessary condition for the effectiveness of an intervention or test; from the patient perspective, if a test is considered acceptable, patients are more likely to adhere to the proposed investigation which in turn results in improved clinical outcomes^{19,20}." This is a qualitative interview study which used a framework approach for data analysis. This is a widely used and accepted research design, particularly in applied health research. We have now tried to clarify our approach to data analysis, which we interpreted as the key issue raised by the reviewer. We have adopted the framework method/approach for data analysis; this is a tool that allows for both inductive and deductive approaches (Gale et al, ref 33). In this study, we adopted an inductive approach, also as described by Gale et al: themes were generated from the data through open coding, and were

	then refined taking into account our research question and adopted definition of acceptability. This information has been added to Methods, Data analysis, paragraph 1: “ Verbatim transcripts were checked, anonymised and analysed using framework analysis³³. This allows for both inductive and deductive approaches. An inductive approach was adopted for data analysis, influenced by our research question and specific definitions of acceptability³³.” We have also moved information regarding the theoretical models used into a new Box ‘Theoretical Models’ found after the Methods, Data analysis section. We have added information about what is included in the models, and provided references that have detailed descriptions for each of them. “Box 1. Theoretical models Three theoretical models, listed below, were used to help inform and conceptualize the themes during data analysis. These models were chosen to underpin important complementary but distinctive aspects of the cancer diagnostic pathway^{35,36} including the role of patient factors/characteristics^{35,36}, the construct of acceptability¹⁹ and patient’s experience of care²⁸. 1. The Model of Pathways to Treatment^{35,36} describes events, processes, intervals and contributing factors (such as patient experience) underpinning the pathway to diagnosing cancer for patients presenting with symptoms. 2. The Theoretical Framework of Acceptability (TFA)¹⁹ describes seven constructs of acceptability: affective attitude, burden, perceived effectiveness, ethicality, intervention coherence, opportunity costs, and self-efficacy. 3. Forster’s et al considerations when assessing acceptability of diagnostic tests²⁸ describes dimensions such as patient-centred care, continuity and coordination of care, and waiting times.”
Results It is better now, thank you. Still, how is the deductive perspective visible?	Please see comment above
Discussion It is stated here that three well-established theoretical models were used but I don’t see how. The authors state that the models complemented each other but how is that shown to the reader? Please see under method. I would like to see a more detailed description of	As per the comments above, we would prefer not to focus too much on the theory as this was used for a specific step of the analysis to help conceptualise the themes, rather than driving the overall analysis. We have provided references with comprehensive descriptions of the constructs within the TFA (reference 19, Sekhon et al) and definitions for care experience

the models to be able to follow the discussion and to grasp the authors statements.	(reference 28, Forster et al) that can now be found in Box 1.
Discussion What about the section on COVID? How is it going to be addressed?	As before, we thought carefully about whether to add findings related to COVID within the main results section. We found that whilst COVID was an important factor in how patients presented to their GP, it did not appear to have an impact on acceptability and experience of using FIT. However, we do consider it is important as background information and for this reason it was included as supplementary data to provide context to the study. We believe that including it within the main results section would deviate from the aims of the study and distract the reader from its main messages. Therefore, we would prefer to keep it as supplementary data. We trust this is appropriate.
Discussion Do you have such solid evidence as to be able to come up with all those recommendations or are they based on other research as well? I'm hesitant.	We do believe that the recommendations are either informed by the results or link clearly to the UK healthcare setting. We have highlighted this further by adding additional information within the Discussion, Implications for research, practice and policy, paragraph 1. “Building on our findings and published literature, alongside considerations on what is feasible within the UK context, we have developed recommendations for research, practice and policy to improve both the FIT experience and patient safety (Box 2).”
Discussion The conclusion needs to be justified in the result section.	We are unsure of which parts of the conclusion are not justified in the results section. Results show that patients find FIT acceptable, that those with previous knowledge or experience of similar tests were happy to do the test, and that despite poor experience of care acceptability was high. Results also show that communication with patients was not always optimal, giving scope for improvement. We are also aware that there are certain groups that our study did not include and we have suggested future studies to include these populations.

VERSION 2 – REVIEW

REVIEWER	Jervaeus, Anna Karolinska Inst
REVIEW RETURNED	03-May-2023

GENERAL COMMENTS	Thank you for letting me review this manuscript titled: It is an interesting paper and I find similarities to the research we have conducted here in Sweden. Can we do better? A qualitative study investigating patient experience and acceptability of using the faecal immunochemical test in primary care Background Relevant to reflect upon: Are the relevant concepts and aspects of the study defined here? For whom are you writing this? Do you need to include any more information, for instance regarding colorectal cancer? I would suggest to expand the background based on these questions. What does NICE stand for? Please explain abbreviations throughout the manuscript. It would be good to put describe the health care system in the UK. Help the reader to grasp the context both regarding health care sector and this part of England. Who do you want to address this paper to? I would suggest formulating a clearer rationale. Do we need more of this type of research and why? What is new? Thanks, the background has improved and I'm satisfied with the new version. Aim The aim is to explore FIT experience and acceptability. This is still not addressed in connection to the aim: How is acceptability defined? How is acceptability addressed in the interview guide? Please see further under Methods. Methods It's good that COREQ is acknowledged. My major concern here is that a deductive analysis seems to have been conducted. That is not, however, visible in the design description. It is not clear how the analysis process has been conducted so I would like to see a more detailed description. It could be in the format of a visual presentation about the process, adding on to the text. New comment: This is still a concern to me, what is the design of this study? To me, a qualitative study is not enough as a description of the design. I would also like to see a rationale for the use of Acceptability much earlier in the manuscript, preferably in the background. Why is this concept relevant and what is included in these conceptual models. Results "Many", "several", "most" participants are frequently used. Is this relevant for a qualitative paper? Isn't it the variation that should be in focus? I would suggest decreasing number of citations in the result section. My impression now is that the section is very much a description of the different citations and that is not my view on how a qualitative result should be structured. It is better now, thank you. Still, how is the deductive perspective visible? Discussion It is stated here that three well-established theoretical models were used but I don't see how. The authors state that the models complemented each other but how is that shown to the reader? Please see under method. I would like to see a more detailed description of the models to be able to follow the discussion and to grasp the authors' statements.
---

	What about the section on COVID? How is it going to be addressed? Do you have such solid evidence as to be able to come up with all those recommendations or are they based on other research as well? Im hesitant. The conclusion needs to be justified in the result section.
--	--

VERSION 2 – AUTHOR RESPONSE

Reviewer comments	Author response
Reviewer 1	
Aim The aim is to explore FIT experience and acceptability. This is still not addressed in connection to the aim: How is acceptability defined? How is acceptability addressed in the interview guide? Please see further under Methods	Following our previous revisions, we have now moved our definition of acceptability into the Introduction (paragraph 4). “Acceptability is defined as a multi-faceted construct comprising affective attitude, burden, perceived effectiveness, ethicality, intervention coherence, opportunity costs, and self-efficacy¹⁹” The questions in the interview topic guide were broad to encompass constructs within Sekhon et al’s theoretical framework of acceptability (TFA). We have added our response here again for information: Within the interview topic guide, questions were broad to both encompass constructs of Sekhon’s TFA, whilst also leaving scope for participants to talk about other issues regarding FIT that they felt were important. For example, affective attitude was investigated by asking questions such as “How did you feel when the GP suggested having a FIT-poo test?” and “How did you feel following the test?”. Burden was explored through questions such as “How did you feel about doing the test – physically, emotionally?”. Intervention coherence was investigated by asking “How would you explain the test to someone else?” The question “How was the FIT-poo test introduced and explained” was designed to shed light on both the care experience and the construct of perceived effectiveness. We did not have questions focusing specifically on ethicality and opportunity costs – while answers to open-ended questions could have still approached issues relevant to these, we found that these constructs were less relevant to the participants’ experiences.
Methods My major concern here is that a deductive analysis seems to have been conducted. That is not, however, visible in the design description. It	Regarding addressing the rationale for the use of acceptability earlier on in the manuscript, we have now added a definition to the Introduction, see above. Furthermore, we have brief

is not clear how the analysis process has been conducted so I would like to see a more detailed description. It could be in the format of a visual presentation about the process, adding on to the text.

New comment: This is still a concern to me, what is the design of this study? To me, a qualitative study is not enough as a description of the design.

I would also like to see a rationale for the use of Acceptability much earlier in the manuscript, preferably in the background. Why is this concept relevant and what is included in these conceptual models.

explained why the concept is relevant (Introduction, paragraph 4):

"It is a necessary condition for the effectiveness of an intervention or test; from the patient perspective, if a test is considered acceptable, patients are more likely to adhere to the proposed investigation which in turn results in improved clinical outcomes^{19,20}."

This is a qualitative interview study which used a framework approach for data analysis. This is a widely used and accepted research design, particularly in applied health research. We have now tried to clarify our approach to data analysis, which we interpreted as the key issue raised by the reviewer.

We have adopted the framework method/approach for data analysis; this is a tool that allows for both inductive and deductive approaches (Gale et al, ref 33). In this study, we adopted an inductive approach, also as described by Gale et al: themes were generated from the data through open coding, and were then refined taking into account our research question and adopted definition of acceptability. This information has been added to Methods, Data analysis, paragraph 1:

"Verbatim transcripts were checked, anonymised and analysed using framework analysis³³. This allows for both inductive and deductive approaches. An inductive approach was adopted for data analysis, influenced by our research question and specific definitions of acceptability³³."

We have also moved information regarding the theoretical models used into a new Box 'Theoretical Models' found after the Methods, Data analysis section. We have added information about what is included in the models, and provided references that have detailed descriptions for each of them.

"Box 1. Theoretical models

Three theoretical models, listed below, were used to help inform and conceptualize the themes during data analysis. These models were chosen to underpin important complementary but distinctive aspects of the cancer diagnostic pathway^{35,36} including the role of patient factors/characteristics^{35,36}, the construct of acceptability¹⁹ and patient's experience of care²⁸.

1. The Model of Pathways to Treatment^{35,36} describes events, processes, intervals and

	contributing factors (such as patient experience) underpinning the pathway to diagnosing cancer for patients presenting with symptoms. 2. The Theoretical Framework of Acceptability (TFA)¹⁹ describes seven constructs of acceptability: affective attitude, burden, perceived effectiveness, ethicality, intervention coherence, opportunity costs, and self-efficacy. 3. Forster's et al considerations when assessing acceptability of diagnostic tests²⁸ describes dimensions such as patient-centred care, continuity and coordination of care, and waiting times."
Results It is better now, thank you. Still, how is the deductive perspective visible?	Please see comment above
Discussion It is stated here that three well-established theoretical models were used but I don't see how. The authors state that the models complemented each other but how is that shown to the reader? Please see under method. I would like to see a more detailed description of the models to be able to follow the discussion and to grasp the authors statements.	As per the comments above, we would prefer not to focus too much on the theory as this was used for a specific step of the analysis to help conceptualise the themes, rather than driving the overall analysis. We have provided references with comprehensive descriptions of the constructs within the TFA (reference 19, Sekhon et al) and definitions for care experience (reference 28, Forster et al) that can now be found in Box 1.
Discussion What about the section on COVID? How is it going to be addressed?	As before, we thought carefully about whether to add findings related to COVID within the main results section. We found that whilst COVID was an important factor in how patients presented to their GP, it did not appear to have an impact on acceptability and experience of using FIT. However, we do consider it is important as background information and for this reason it was included as supplementary data to provide context to the study. We believe that including it within the main results section would deviate from the aims of the study and distract the reader from its main messages. Therefore, we would prefer to keep it as supplementary data. We trust this is appropriate.
Discussion Do you have such solid evidence as to be able to come up with all those recommendations or are they based on other research as well? I'm hesitant.	We do believe that the recommendations are either informed by the results or link clearly to the UK healthcare setting. We have highlighted this further by adding additional information within the Discussion, Implications for research, practice and policy, paragraph 1. "Building on our findings and published literature, alongside considerations on what is feasible within the UK context, we have developed recommendations for research,

	practice and policy to improve both the FIT experience and patient safety (Box 2)."
Discussion The conclusion needs to be justified in the result section.	We are unsure of which parts of the conclusion are not justified in the results section. Results show that patients find FIT acceptable, that those with previous knowledge or experience of similar tests were happy to do the test, and that despite poor experience of care acceptability was high. Results also show that communication with patients was not always optimal, giving scope for improvement. We are also aware that there are certain groups that our study did not include and we have suggested future studies to include these populations.

VERSION 3 – REVIEW

REVIEWER	Jervaeus, Anna Karolinska Inst
REVIEW RETURNED	30-May-2023

GENERAL COMMENTS	Thank you for letting me review this manuscript titled: It is an interesting paper and I find similarities to the research we have conducted here in Sweden. Can we do better? A qualitative study investigating patient experience and acceptability of using the faecal immunochemical test in primary care Background Relevant to reflect upon: Are the relevant concepts and aspects of the study defined here? For whom are you writing this? Do you need to include any more information, for instance regarding colorectal cancer? I would suggest to expand the background based on these questions. What does NICE stand for? Please explain abbreviations throughout the manuscript. It would be good to put describe the health care system in the UK. Help the reader to grasp the context both regarding health care sector and this part of England. Who do you want to address this paper to? I would suggest formulating a clearer rationale. Do we need more of this type of research and why? What is new? Thanks, the background has improved and I'm satisfied with the new version. Aim The aim is to explore FIT experience and acceptability.
---

This is still not addressed in connection to the aim: How is acceptability defined? How is acceptability addressed in the interview guide? Please see further under Methods.

230530: Ok better now.

Methods

It's good that COREQ is acknowledged.

My major concern here is that a deductive analysis seems to have been conducted. That is not, however, visible in the design description. It is not clear how the analysis process has been conducted so I would like to see a more detailed description. It could be in the format of a visual presentation about the process, adding on to the text.

New comment: This is still a concern to me, what is the design of this study? To me, a qualitative study is not enough as a description of the design.

I would also like to see a rationale for the use of Acceptability much earlier in the manuscript, preferably in the background. Why is this concept relevant and what is included in these conceptual models

230530: Ok now.

Results

"Many", "several", "most" participants are frequently used. Is this relevant for a qualitative paper? Isn't it the variation that should be in focus? I would suggest decreasing number of citations in the result section. My impression now is that the section is very much a description of the different citations and that is not my view on how a qualitative result should be structured.

It is better now, thank you. Still, how is the deductive perspective visible?

Discussion

It is stated here that three well-established theoretical models were used but I don't see how. The authors state that the models complemented each other but how is that shown to the reader? Please see under method. I would like to see a more detailed description of the models to be able to follow the discussion and to grasp the authors' statements.

What about the section on COVID? How is it going to be addressed?

Do you have such solid evidence as to be able to come up with all those recommendations or are they based on other research as well? I'm hesitant.

The conclusion needs to be justified in the result section.

230530: Ok